**PLOS** | **ONE**

# Trends and risk factors associated with stillbirths: A case study of the Navrongo War Memorial Hospital in Northern Ghana

**Engelbert A. Nonterah**[1,2,3]*, **Isaiah A. Agorinya**[1,4], **Edmund W. Kanmiki**[5], **Juliana Kagura**[6], **Mariatu Tamimu**[7], **Emmanuel Y. Ayamba**[1], **Esmond W. Nonterah**[1], **Michael B. Kaburise**[1,2], **Majeedallahi Al-Hassan**[2], **Winfred Ofosu**[8], **Abraham R. Oduro**[1], **John K. Awonoor-Williams**[9]

**1** Navrongo Health Research Centre, Ghana Health Service, Navrongo, Ghana, **2** Navrongo War Memorial Hospital, Ghana Health Service, Navrongo, Ghana, **3** Julius Global Health, Julius Centre for Health Sciences and Primary Care, University Medical Centre Utrecht, Utrecht University, Utrecht, the Netherlands, **4** Swiss Tropical and Public Health Institute, Socinstrasse, Basel, Switzerland, University of Basel, Peterplatz, Basel, Switzerland, **5** Regional Institute for Population Studies (RIPS), University of Ghana, Legon, Accra, Ghana, **6** Department of Epidemiology and Biostatistics, School of Public Health, University of the Witwatersrand, Faculty of Health Sciences, Johannesburg, South Africa, **7** Obstetrics and Gynaecology Department, University of Nairobi, Nairobi, Kenya, **8** Upper East Regional Health Directorate, Ghana Health Service, PMB, Bolgatanga, Ghana, **9** Policy, Planning Monitoring Division, Ghana Health Service, Headquarters, Accra, Ghana

* drenanonterah@gmail.com

**Data Availability Statement:** All relevant data are within the manuscript and its Supporting Information files.

## Abstract

### Background

Maternal and Child health remains at the core of global health priorities transcending the Millennium Development Goals into the current era of Sustainable Development Goals. Most low and middle-income countries including Ghana are yet to achieve the required levels of reduction in child and maternal mortality. This paper analysed the trends and the associated risk factors of stillbirths in a district hospital located in an impoverished and remote region of Ghana.

### Methods

Retrospective hospital maternal records on all deliveries conducted in the Navrongo War Memorial hospital from 2003–2013 were retrieved and analysed. Descriptive and inferential statistics were used to summarise trends in stillbirths while the generalized linear estimation logistic regression is used to determine socio-demographic, maternal and neonatal factors associated with stillbirths.

### Results

A total of 16,670 deliveries were analysed over the study period. Stillbirth rate was 3.4% of all births. There was an overall decline in stillbirth rate over the study period as stillbirths declined from 4.2% in 2003 to 2.1% in 2013. Female neonates were less likely to be stillborn (Adjusted Odds ratio = 0.62 and 95%CI [0.46, 0.84]; p = 0.002) compared to male neonates;

**Funding:** The authors received no specific funding for this work.

**Competing interests:** The authors have declared that no competing interests exist.

neonates with low birth weight (4.02 [2.92, 5.53]) and extreme low birth weight (18.9 [10.9, 32.4]) were at a higher risk of still birth (p<0.001). Mothers who had undergone Female Genital Mutilation had 47% (1.47 [1.04, 2.09]) increase odds of having a stillbirth compared to non FGM mothers (p = 0.031). Mothers giving birth for the first time also had a 40% increase odds of having a stillbirth compared to those who had more than one previous births (p = 0.037).

## Conclusion

Despite the modest reduction in stillbirth rates over the study period, it is evident from the results that stillbirth rate is still relatively high. Primiparous women and preterm deliveries leading to low birth weight are identified factors that result in increased stillbirths. Efforts aimed at impacting on stillbirths should include the elimination of outmoded cultural practices such as FGM.

## Background

Stillbirths are among the most common adverse pregnancy outcomes with accompanying long-lasting deep psychological effects on parents, care givers, health providers and the community at large [1–3]. Globally, over 2.6 million babies are stillborn annually and 98% of all these stillbirths are said to be occurring in low and middle income countries [4].

Due to the huge differences in prevalence of stillbirth between high and Low to Middle Income Countries (LMIC), stillbirth equally used as a global development indicator [5]. Stillbirth has a direct association with availability of obstetric care services and so it disproportionately affects the poor in resource-constraint countries more than high-resource countries [6] with a reported prevalence of 55% among rural families [7]. This pattern of distribution mirrors maternal deaths and correlates mainly with areas of low skilled birth attendants (SBA), inadequate health infrastructure and lack of emergency obstetric care service [8]. Currently, 77% of the 98% stillbirths recorded in LMIC occur mainly in South-East Asia and sub-Saharan Africa (SSA) [4, 8]. In SSA particularly, women are reported to be 24 and 50 times more at risk of having stillbirths during labour than women in North America and United Kingdom respectively [6, 9]. The Ghana Maternal Health Survey 2017 observed 68% of stillbirths occur in rural areas compared to urban areas [10]. Saleem et al (2018) observed a 3.0% annual decline in stillbirth rate globally with much lower rates of decline for countries in South-East Asia and Africa [9]. This the author observed will further result in the inability of these countries to achieve the Every Newborn Action Plan goal of 12 per 1000 births by the year 2030 [9].

The slow pace of decline in resource-constraint settings can be attributed to the lack of SBAs, inadequate emergency obstetric services and inadequate health infrastructure. For instance, districts hospitals in Ghana have an average of one Medical Officer at post, yet emergency Obstetric and Gynaecological surgeries have been previously reported to represent a substantial proportion of the clinical duties of Medical Officers. Particularly, obstetric complications often requiring caesarean section are the commonest cases encountered [11].

In addressing these challenges, Ghana, recently implemented a number of maternal and child health related initiatives with the core objective of improving universal access to comprehensive obstetric care through an increase in institutional deliveries, where births are attended to by SBAs and improving obstetric referrals [12]. The Upper East Region of Ghana has since

seen a steady improvement in maternal and child health care service provision. Notable improvements include increase in institutional deliveries from 46% in 2008 to 77% in 2010 and improved obstetric and new-born referrals at all four levels of the health system: Community-based Health and Planning Services compounds to Health Centres through the District Hospitals and finally to the Regional Hospital [13].

However to date, little is documented about how the observed improvements in obstetric care services has impacted on pregnancy outcomes in the region. Few longitudinal studies exist in Ghana and SSA on stillbirth as a monitoring tool for improving obstetric care. Most of the epidemiological studies assessing pregnancy outcomes use perinatal mortality which combines both stillbirths and early neonatal deaths and are often based on complex statistical modelling methods, country-wide or community demographic health surveillance and community verbal autopsies [5]. The value of these estimates is limited because of etiological differences between stillbirths and early neonatal deaths [12, 14], especially that labour and delivery (intrapartum period) are the highest risk periods for stillbirths [15–18]. A major challenge to preventing or reducing the incidence of stillbirths is the paucity of context specific knowledge on the causes and risk factors associated with stillbirths especially in resource poor settings [2]. This paper provides a ten-year longitudinal analysis of the trend in stillbirth rate and examines the neonatal and maternal characteristics associated with stillbirths using data from a hospital located in a rural poor setting in northern Ghana.

## Materials and methods

### Study setting

This study used data collected from the Navrongo War Memorial Hospital (WMH) in the Kassena-Nankana Municipality of the Upper East Region of Northern Ghana. The WMH is located in Navrongo and is the only secondary referral facility in the Kassena-Nankana area offering emergency obstetric care, surgical and other public health services [19]. It is a 123 bed facility serving a population of about 165,000 people and receives referrals from the many primary level health facilities in the area including private clinics as well as referrals from neighbouring districts and neighbouring towns in Burkina Faso [20]. Unpublished data from the hospital shows an average yearly Out Patient Department attendance of about 55,000 with an average of 10,000 in-patient admissions of which maternal related conditions form part of the top ten causes of admissions. The annual Antenatal attendance is about 2,500 and annual deliveries are hovering around 1,500.

The Kassena-Nankana Municipality which is home to the WMH lies within the Guinea Savannah ecological zone in the extreme north-eastern part of Ghana and occupies an area of about 1,675 square kilometres. It borders the Bulisa and Sissala Districts to the South West and West respectively, Bongo and Bolgatanga Districts in East and North-East respectively, and Burkina Faso in the North-eastern corner. About 90% of the population live in rural communities, with a small suburban area within the capital town, Navrongo. Settlements are mainly sparse with closely knit extended families living in the same compound with average of 10 people per compound. There are two main climatic seasons: A short rainy season between June and September and a dry season between October and May with the harmattan winds peaking in January-February. The average rainfall is between 850-1000mm while temperatures range from 20˚C to as high as 40˚C during the dry season. The vast majority of residents in the Upper East region are engaged in subsistence rain-fed agriculture. However, over-cropping and increasingly erratic rainfall have diminished agricultural productivity, exacerbating pervasive poverty [21, 22]. As a consequence of these circumstances, the region ranks among Ghana's three most impoverished regions with a poverty prevalence of 55% [21].

## Data collection

Delivery records data was retrospectively assembled from the Navrongo WMH maternity ward record books. The hospital keeps record books that contain systematically documented information on each patient attending the hospital. These hospital record books includes a maternal register that is used to capture information on all maternal related cases, information captured includes maternal demographic characteristics, mode of delivery, delivery outcome (live, fresh and macerated stillbirths and birth weight), gestation at delivery, genital mutilation status of the mother, relevant previous obstetric history etc. A structured designed data capturing tool was used to extract all information needed for the study from delivery record books in the maternity ward. Extracted data span a ten year period (2003–2013). The information was extracted by four trained research assistants independently to eliminate bias and errors and to ensure reliability. The extracted data was vetted and confirmed by two physicians independently before data was entered electronically for analysis.

## Data analysis

The main outcome of interest in this study was stillbirth. We applied the standard definition of stillbirth being any foetal loss after 28 weeks of gestation, and or a weight of 1000g (1kg) [4, 23]. The ICD-10 further states the foetus must be 35cm of length or more to qualify as a stillbirth [23]. We however had no information on the foetal length at birth. We also categorised Birth weight less than 2.5Kg as Low Birth weight (LBW) as recommended by WHO [24]. A fresh stillbirth was defined as the intrauterine death of a foetus during labour or delivery, and a macerated stillbirth was defined as the intrauterine death of a foetus before the onset of labour or 24 hours before delivery, where the foetus showed degenerative changes [24] as reported in the obstetric records by the attending physician or midwife. Primiparous woman refers to a pregnant woman who had not delivered before, with the index pregnancy being the first expected delivery. Multiparous woman has more than 1 delivery but less than five and a grand multiparous woman has greater than five deliveries. A termed pregnancy is defined as 37 completed weeks of gestation to 40 weeks while preterm pregnancy is defined as one less than 37 weeks gestation and post term pregnancy is greater than 40weeks gestation. The proportions of post term deliveries were so negligible and therefore had no statistical significance so were added these to term deliveries for the analysis. The data were entered in Epidata 3.1 and exported to STATA version 14.2 (Statacorp College Station, Texas 77845 USA) for analysis. Descriptive analyses were first used to summarise maternal and new-born characteristics such as maternal age, type of delivery, delivery outcome, sex of neonate, perineum, use of partograph, birth weight, gestational age, maternal parity and Female Genital Mutilation (FGM) status of mother. In addition, trends in stillbirth rate were presented over the ten year period using a line graph and reported as absolute counts with respective proportions. Next, we performed bivariate analysis of these maternal and neonatal factors with delivery outcome (either live birth or stillbirth) and report examined differences using Pearson's Chi Squared ($\chi^2$) test.

To determine the maternal and neonatal factors associated with stillbirths, we employed the Generalized Estimation Equation (GEE) regression models. The GEE formulated by Liang and Zeger (1986) was found to be the best regression approach for estimating the relationship of factors associated with stillbirth because of the correlation in our data which is an important assumption for using GEE [25]. In addition the GEE is said to use the generalized linear regression models to provide more unbiased and efficient parameters as compared to the ordinary least squares approach because it permits specification of correlation matrix accounting for within subject correlation of responses on dependent variables of different distributions [26]. Variables from the bivariate analyses that were significant at a 10% significance level were

included in the GEE regression models. Associations from the GEE are presented as unadjusted and adjusted odds ratios (AOR) with corresponding 95% confidence intervals and significance set at two-tailed $p<0.05$.

### Ethical considerations

The Navrongo WMH administration and the Upper East Regional Health Directorate of the Ghana Health Service granted approval for the study. Additional ethics approval was obtained from the Navrongo Health Research Centres Institutional Review Board. Because secondary data was used, informed consent was waived. Further to this, we ensured confidentiality and anonymity in data extraction, processing/cleaning and analyses.

## Results

### Background characteristics of study participants

Presented in Table 1 are background characteristics of study participants.

A total of 16,670 maternal deliveries were recorded in the WMH over the study period out of which 53.8% of the neonates were males and 46.2% females. Out of the total deliveries, 96.6% were live births and 3.5% were stillbirths. Of the 565 stillbirths recorded, 2.2% were fresh stillbirths and 1.2% macerated stillbirths. Average maternal age was 26 years with 74.1% of mothers in the 20-34years age group and advanced maternal age of 35+ years being the least at 10.8% (see Table 1). About 15.1% of the participants were less than 20 years denoting teenage pregnancy.

Most deliveries were from multiparous women representing 58.8% followed by grand multiparous women with 39.3%, and the least was primiparous women (1.9%). The predominant mode of delivery was by spontaneous vagina delivery (82.6%) followed by caesarean section (12.5%) and assisted or instrumental deliveries (5.0%). Preterm deliveries were 60.3%, and term deliveries were 39.7%. Approximately 42.4% of mothers were unemployed and 5.1% were students. Of the employed group 5.7% were Government employees, 36.3% were self-employed engaging mostly in petty trading while 7.5% had farming as their main occupation.

About 69% of expectant mothers were monitored with the partograph while 31.0% were not. Of all the deliveries documented 18% of the mothers had undergone FGM. A total of 4,876 women were tested for Human Immunodeficiency Virus (HIV) as part of Prevention of Mother-To-Child Transmission, out of which 1.3% tested positive.

### Trends in stillbirth rate

Fig 1 presents the trends in stillbirth rates from 2003 to 20013. Results suggest a general decline in trends of stillbirth over the entire 10 year period.

The overall stillbirth rate was 4.2% in 2003 and this declined to 2.1% in 2013. Fresh stillbirths as at 2003 represented 2.8% and declined to 1.3% in 2013, and macerated stillbirths declined from 1.4% to 0.7% 2013 [Fig 1]. The highest rate of decline in stillbirth rate recorded was from 2010 (3.6%) to 2013 (2.1%), a period during which, institutional (hospital) deliveries were on the rise.

### Bivariate and multivariate analysis of stillbirths

Results of bivariate analysis of the association of selected foetal and maternal factors with stillbirth status are presented in Table 2. At the bivariate level all neonatal and maternal factors had varied associations with stillbirth in the study population.

**Table 1. Maternal and New-born background characteristics.**

| Characteristics | Number (n) | Percentage (%) | 95% CI |
|---|---|---|---|
| **Type of delivery** | | | |
| Spontaneous Vaginal Delivery | 13752 | 82.5 | 81.9, 83.1 |
| Caesarean-Section | 2080 | 12.5 | 12.0, 13.0 |
| Assisted/Instrumental | 838 | 5.0 | 4.7, 5.4 |
| **Birth outcome** | | | |
| Single Birth | 16103 | 97.2 | 96.9, 97.4 |
| Multiple Birth | 468 | 2.8 | 2.6, 3.1 |
| **Sex of neonate** | | | |
| Male | 8816 | 53.8 | 53.0, 54.6 |
| Female | 7572 | 46.2 | 45.4, 47.0 |
| **Delivery outcome** | | | |
| Alive birth | 16084 | 96.6 | 96.3, 96.9 |
| Fresh Stillbirth | 373 | 2.2 | 2.0, 2.5 |
| Macerated Stillbirth | 192 | 1.2 | 1.0, 1.3 |
| **State of perineum** | | | |
| Intact | 3198 | 47.4 | 46.2, 48.6 |
| Laceration | 1924 | 28.5 | 27.4, 29.6 |
| Episiotomy | 1628 | 24.1 | 23.1, 25.1 |
| **Use of Partograph** | | | |
| Yes | 3280 | 69.0 | 67.6, 70.3 |
| No | 1472 | 30.9 | 29.6, 32.3 |
| **Birth weight (Kg)** | | | |
| Extreme LBW (<1.5) | 248 | 1.5 | 1.3, 1.7 |
| Low birth weight (>1.5–2.5) | 2385 | 14.5 | 13.9, 15.0 |
| Normal birth weight (>2.5) | 13831 | 84.0 | 83.4, 84.6 |
| **Estimated Gestational age (weeks)** | | | |
| Preterm (<37 weeks) | 8942 | 60.3 | 59.5, 61.1 |
| Term pregnancy (38–40 weeks) | 5615 | 37.9 | 37.1, 38.7 |
| Post-term (> 40weeks) | 264 | 1.8 | 1.6, 2.0 |
| **Maternal Age (years)** | | | |
| < 20 years | 2444 | 15.1 | 14.6, 5.7 |
| 20–34 years | 11836 | 74.1 | 73.4, 74.7 |
| > 35 years | 1720 | 10.8 | 10.3, 11.3 |
| **Maternal Parity** | | | |
| Primiparous | 313 | 1.9 | 1.7, 2.1 |
| Multiparous | 15374 | 92.2 | 91.8, 92.6 |
| Grand multiparous | 983 | 5.9 | 5.5, 6.3 |
| **Female Genital Mutilation** | | | |
| Yes | 1646 | 18.6 | 17.8, 19.4 |
| No | 7201 | 81.4 | 80.6, 82.2 |

For instance, male neonates (p = 0.008), mothers older than 35 years (p = 0.021), nulliparous women (p<0.0001), unemployed women (p<0.0001), neonates born preterm (p<0.0001) and extreme LBW neonates (p<0.0001) were more likely to be born still than the rest. More caesarean deliveries also had a trend towards stillbirth (p = 0.098) compared to SVDs.

Presented in Table 3 are the GEE regression analyses showing factors associated stillbirth.

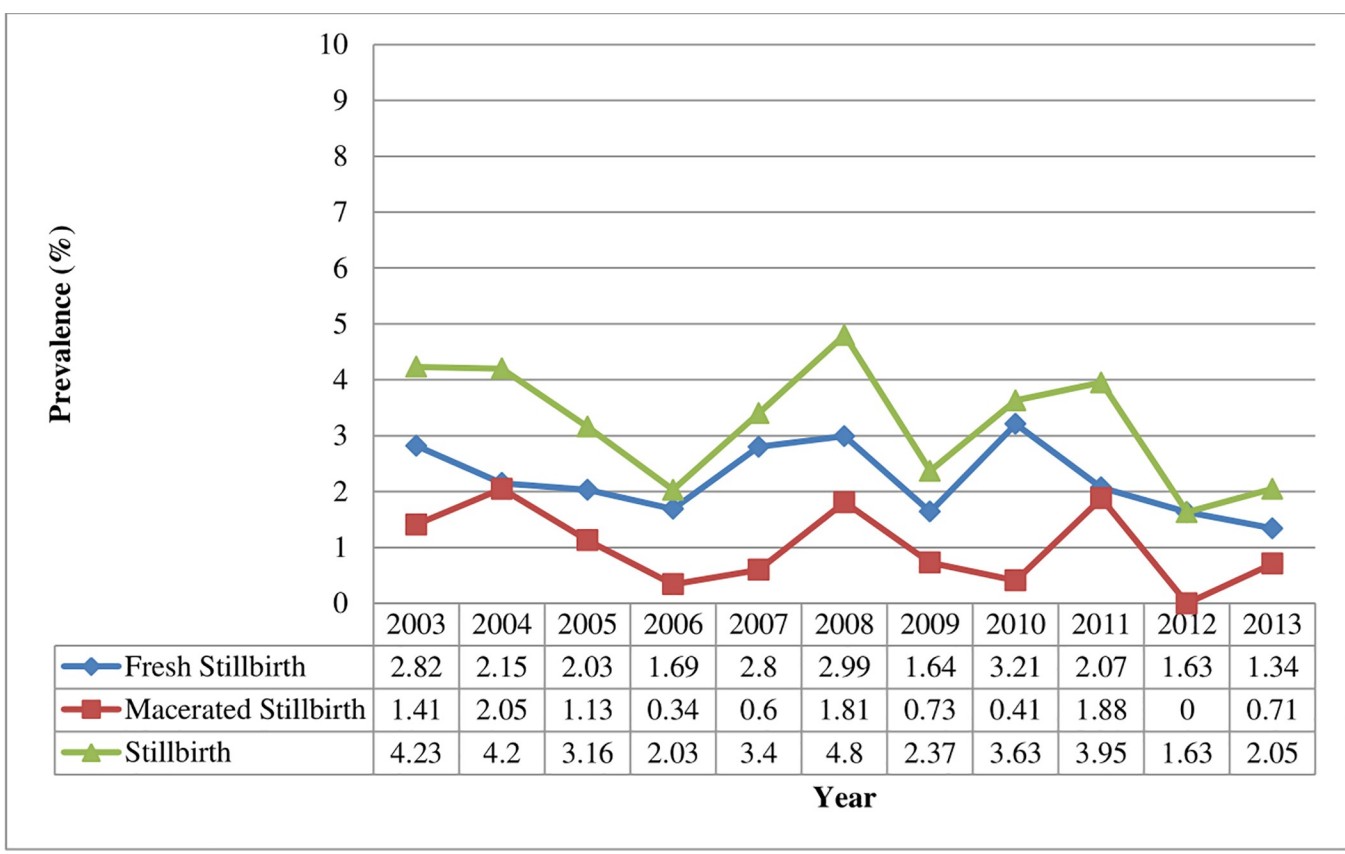

**Fig 1. Trends in stillbirth rates from 2003–2013 expressed as percentage of the number of births per year.**

Female neonates had a lower risk of being stillborn compared to their male counterparts (Adjusted Odds Ratio, AOR = 0.62 95%CI [0.46, 0.84]); p = 0.002]. Although maternal age was significantly associated with delivery outcome in bivariate analysis and the unadjusted model, it was not significant in the adjusted regression analysis. Also maternal occupation turn out not to be significantly associated with birth outcome in multivariate analysis.

Neonates were extreme LBW (AOR = 18.9 [10.9, 32.4]) and LBW (AOR = 4.02 [2.92, 5.53]) were at a higher odds of been stillborn than those with normal weight.

Mothers who had some form of FGM had 47% odds of having stillbirths compared to those who have not undergone FMG (AOR = 1.47 [1.04, 2.09]). Nulliparous women were at a higher odds of having stillbirths (AOR = 1.40 [0.73, 2.67]) during delivery compared to multiparous women. On the contrary, grand multiparous women were however, less likely to have stillbirths (AOR = 0.65 [0.46, 0.92]). Gestational age of foetus was not significantly associated with stillbirth experience in multivariate analysis.

## Discussion

Still births is a major problem in developing countries were about 98% of the incidence of stillbirths are said be occurring [4]. To achieve the required reductions in the global burden of stillbirths, proven strategic interventions that are anchored on a clear understanding of the causes and factors associated with the incidence of stillbirths are required [27]. However, there is paucity of such context specific evidence in most rural poor settings. In this study, we assess the trends in stillbirth rates over a ten year period and examine the maternal and neonatal

**Table 2. Bivariate analysis of stillbirth and maternal and new born characteristics.**

| Maternal and foetal characteristics | Number of births | Proportion of Live births (%) | Proportion of Stillbirths (%) | P-value |
|---|---|---|---|---|
| **Sex of neonate** | | | | |
| Male | 8781 | 8544 (97.3) | 237 (2.7) | 0.008 |
| Female | 7553 | 7397 (97.9) | 156 (2.0) | |
| **Maternal age (years)** | | | | |
| 12–19 | 2514 | 2444 (97.2) | 70 (2.8) | 0.021 |
| 20–34 | 12253 | 11836 (96.6) | 417 (3.4) | |
| 35+ | 1798 | 1720 (95.7) | 78 (4.3) | |
| **Maternal parity** | | | | <0.0001 |
| Primiparous | 313 | 289 (92.3) | 24 (7.7) | |
| Multiparous | 9704 | 9362 (96.5) | 342 (3.5) | |
| Grand multiparous | 6512 | 6317 (97.0) | 195 (3.0) | |
| **Maternal Occupation** | | | | |
| Unemployed | 6874 | 6553 (95.3) | 321 (4.7) | <0.0001 |
| Farmer | 1225 | 1188 (97.5) | 37 (3.0) | |
| Self-employed | 5880 | 5735 (97.5) | 145 (2.5) | |
| Student | 833 | 815 (97.8) | 18 (2.1) | |
| Government employed | 933 | 915 (98.1) | 18 (1.9) | |
| Other | 434 | 420 (96.8) | 14 (3.2) | |
| **Gestation** | | | | |
| Preterm | 8876 | 8530 (96.1) | 346 (3.9) | <0.0001 |
| Term | 5571 | 5435 (97.6) | 136 (2.4) | |
| Post-term | 261 | 257 (98.5) | 4 (1.5) | |
| **Birth weight** | | | | |
| Extreme LBW | 243 | 179 (73.7) | 64 (26.4) | |
| LBW | 2369 | 2216 (93.5) | 153 (6.5) | <0.0001 |
| Normal birth weight | 13729 | 13413 (97.7) | 316 (2.3) | |
| **Type of Delivery** | | | | 0.098 |
| SVD | 13651 | 13215 (96.8) | 436 (3.1) | |
| Caesarean Section | 2059 | 1975 (95.2) | 84 (4.1) | |
| Assisted/Instrumental delivery | 835 | 810 (97.0) | 25 (3.0) | |

factors associated with still births in a hospital located in a rural impoverished setting of northern Ghana.

Overall, we found a stillbirth rate of 34 per 1000 live births which is higher than the country rate of 14 per 1000 live births [28] Interestingly there was a decline in stillbirth rate from 42 per 1000 live births in 2003 to 21 per 1000 live births by the year 2013. This modest achievement notwithstanding, the reported rate remains high. However, these findings are lower than what has been reported in other developing countries like Nigeria which has an average of 42 per 1000 live births, Pakistan with 47 per 1000 live births and The Gambia with 156 per 1000 live births [7, 29]. General improvements in health care delivery across the Upper East Region [13] is said to have increased access to healthcare services resulting from the ever improving Community-based Health and Planning Services in the region could account for the low SB rate documented. The trends in SB rate over the period indicate an annual rate of decline of 0.2% far lower than the overall reported annual rate of decline for LMIC, high income countries and the global rate [30]. However, the peak of the decline in stillbirths for the study area was from 2010 to 2013 and this period coincides with increased institutional deliveries and

**Table 3. Association between maternal and new born characteristics and stillbirth GEE Model.**

| Risk factors | n | Unadjusted OR (95% CI) | P-value | Adjusted OR (95% CI) | P-value |
|---|---|---|---|---|---|
| **Sex of neonate** | | | | | |
| Male | 8781 | 1 | | **1** | |
| Female | 7553 | 0.76 (0.62, 0.93) | 0.009 | **0.62 (0.46, 0.84)** | **0.002** |
| **Mothers age** | | | | | |
| < 20 years | 2514 | 0.81 (0.63, 1.05) | 0.022 | 0.66 (0.33, 1.29) | 0.270 |
| 20–34 years | 12253 | 1 | | 1 | |
| ≥ 35 years | 1798 | 1. 28 (1.01, 1.65) | | 1.04 (0.66, 1.65) | |
| **Mother Occupation** | | | | | |
| Unemployed | 6874 | 1 | | 1 | |
| Farmer | 1225 | 0.64 (0.45, 0.89) | <0.0001 | 0.82 (0.39, 1.71) | 0.280 |
| Self-employed | 5880 | 0.52 (0.42, 0.63) | | 0.67 (0.47, 0.94) | |
| Student | 833 | 0.45 (0.28, 0.73) | | 1.06 (0.47, 2.39) | |
| Government employed | 933 | 0.40 (0.25, 0.65) | | 0.73 (0.31, 1.72) | |
| Other | 434 | 0.68 (0.39, 1.17) | | 1.19 (0.51, 2.81) | |
| **Birth weight** | | | | | |
| Normal birth weight | 13729 | 1 | | **1** | |
| Extreme LBW | 243 | 15.2 (11.2, 20.6) | <0.0001 | **18.9 (10.9, 32.4)** | **<0.0001** |
| LBW | 2369 | 2.93 (2.40, 3.57) | | **4.02 (2.92, 5.53)** | |
| **FGM status** | | | | | |
| Yes | 7201 | 1 | | **1** | |
| No | 1646 | 1.56 (1.21, 1.99) | 0.001 | **1.47 (1.04, 2.09)** | **0.031** |
| **Parity** | | | | | |
| Multiparous | 9704 | 1 | | **1** | |
| Primiparous | 313 | 2.27 (1.48, 3.49) | <0.0001 | **1.40 (0.73, 2.67)** | **0.037** |
| Grand multiparous | 6512 | 0.85 (0.71, 1.01) | | **0.65 (0.46, 0.92)** | |
| **Gestation** | | | | | |
| Preterm | 8876 | 1.62 (1.33, 1.98) | <0.0001 | 1.19 (0.85, 1.66) | |
| Term | 5571 | 1 | | 1 | 0.301 |
| Post-term | 261 | 0. 62 (0.23, 1.69) | | 0.84 (0.60, 1.17) | |

improved emergency obstetric referrals in the area probably resulting from the implementation of the millennium accelerated framework for the achievement of the MDG 4 and 5 in Ghana.

Logistic regression analysis found that younger and advanced maternal ages did not confer an increased risk to SB and this is consistent with available literature that showed most SBs in Ghana occurred in the 20–29 year group [28]. However, a study in India identified younger and advanced maternal ages as important risk factors to stillbirths [31] suggesting that maternal age as a risk factor for stillbirth may be setting specific. Maternal occupation did not confer a significant risk to stillbirth in this study.

From the results, majority of the deliveries were by Spontaneous Vaginal Delivery followed by C-Section and assisted deliveries consistent with a study conducted in The Gambia which found that about four out of five births were by spontaneous vaginal delivery [29]. The C-section rate over the study period was 12.5% within the 10–15% proposed by the World Health Organization but far lower than the rates reported in the developed world.

Our study showed that three out of five stillbirths were due to preterm deliveries and this is consistent with studies conducted in Burkina Faso [32] and also by the Global Alliance to

prevent stillbirths and prematurity which reported that majority of stillbirths involve preterm foetuses especially the extremely premature [33].

In bivariate analysis, sex of neonate, birth weight, FGM status of mother, maternal age, maternal occupation, gestation age, mode of delivery and parity were significantly associated with the birth outcome. However after adjusting for confounding factors in multivariable analyses, the risk factors for stillbirths in the district were sex of neonate, weight of the neonate, FGM, and parity. The effect of FGM on birth outcomes has previously been documented in the same area which found that babies from circumcised mothers were about two times more likely to be stillborn [19]. This is a significant finding as FGM has been part of the passage rite to adulthood for several decades in the study area [34, 35]. However due to culturally appropriate community interventions embarked upon by the Navrongo Health Research Centre in the past decade, prevalence of FGM has been declining in recent years [19]. This study showed a further drop in the prevalence of FGM from 9.0% to 0.4% over the study period.

We also found that primiparous women had increased risk of stillbirths which was similar to findings reported in Dublin, Ireland [36] and in Ghana [37]. Even though prevalence of HIV infection in pregnancy during the study period was low at 1.3%, it is discouraging to know that only a smaller proportion 28% of expectant mothers were counselled and tested. The study did not find any association between HIV and the risk of having a stillborn as reported by some studies in SSA [38].

Previous studies in Ghana and Angola have identified the use of partograph in monitoring labour as an appropriate tool for improving pregnancy outcomes through early referrals for interventions such as augmentation and emergency surgery when indicated [12, 39]. Our study found over half of the expectant mothers had their labour monitored on the partograph. This improvement can be attributed to continuous training of midwives by Upper East Regional Health Directorate as part of its audit of obstetric and new-born referrals in the region [12].

## Study limitation (s)

The use of hospital based data is limited by deficiency in data collection and collation and as a retrospective study, some important obstetric outcomes and indications may not have been consistently documented and therefore could have not permitted further sub analysis. However, as an index study to assess the true burden of stillbirth and the trends at the facility level, the results do provide significant evidence for the prevailing situation and forms the basis for which future evaluation can be compared.

## Conclusion

The study showed modest decline in stillbirth rate over the study period. It is evident from the study that stillbirth rate is still relatively high. Primiparous women and preterm deliveries leading to low birth weight are identified factors that result in increased stillbirths. Efforts aimed at impacting on stillbirths should include the elimination of outmoded cultural practices such as FGM. Health systems strengthening especially in the utilization of antenatal care services and skilled attendants at delivery are essential in improving foeto-maternal outcomes.

## Supporting information

**S1 Table. Data used for the analyses (Stillbirth data).**
(XLS)

## Acknowledgments

We are very grateful to the mothers and babies whose records were used. The authors are also grateful to the management of the district hospital and the Regional Health Directorate for granting permission for the use of the hospital data. We are especially indebted to staff in the labour ward for doing due diligence with record keeping. We express our gratitude to the following research assistants; Enyonam Duah, Mathilda Tsifodze, Joshua Ti-ire Ang and Philemon Aduntera for the diligent and detailed work with data collection.

## Author Contributions

**Conceptualization:** Engelbert A. Nonterah, Isaiah A. Agorinya, Edmund W. Kanmiki, John K. Awonoor-Williams.

**Data curation:** Engelbert A. Nonterah, Isaiah A. Agorinya, Emmanuel Y. Ayamba, Esmond W. Nonterah.

**Formal analysis:** Engelbert A. Nonterah, Isaiah A. Agorinya.

**Funding acquisition:** Engelbert A. Nonterah.

**Investigation:** Engelbert A. Nonterah, Isaiah A. Agorinya, Edmund W. Kanmiki, Majeedallahi Al-Hassan, Winfred Ofosu, Abraham R. Oduro, John K. Awonoor-Williams.

**Methodology:** Engelbert A. Nonterah, Isaiah A. Agorinya, Edmund W. Kanmiki, John K. Awonoor-Williams.

**Project administration:** Engelbert A. Nonterah, Majeedallahi Al-Hassan, Winfred Ofosu, Abraham R. Oduro, John K. Awonoor-Williams.

**Resources:** Engelbert A. Nonterah, Majeedallahi Al-Hassan, Winfred Ofosu, Abraham R. Oduro, John K. Awonoor-Williams.

**Software:** Engelbert A. Nonterah.

**Supervision:** Engelbert A. Nonterah, Winfred Ofosu, Abraham R. Oduro, John K. Awonoor-Williams.

**Validation:** Engelbert A. Nonterah, Isaiah A. Agorinya, Edmund W. Kanmiki, Juliana Kagura, Mariatu Tamimu, Esmond W. Nonterah, Winfred Ofosu, Abraham R. Oduro, John K. Awonoor-Williams.

**Visualization:** Engelbert A. Nonterah, Isaiah A. Agorinya, Edmund W. Kanmiki, Juliana Kagura, Mariatu Tamimu, Esmond W. Nonterah, Winfred Ofosu, Abraham R. Oduro, John K. Awonoor-Williams.

**Writing – original draft:** Engelbert A. Nonterah, Isaiah A. Agorinya, Edmund W. Kanmiki.

**Writing – review & editing:** Engelbert A. Nonterah, Isaiah A. Agorinya, Edmund W. Kanmiki, Juliana Kagura, Mariatu Tamimu, Emmanuel Y. Ayamba, Esmond W. Nonterah, Michael B. Kaburise, Majeedallahi Al-Hassan, Winfred Ofosu, Abraham R. Oduro, John K. Awonoor-Williams.

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
