## [Decision Letter · Decision Letter 0]

11 Nov 2019

PONE-D-19-24698

Trends and risk factors associated with stillbirths: A case study of the Navrongo War Memorial Hospital in Northern Ghana

PLOS ONE

Dear Dr Nonterah,

Thank you for submitting your manuscript to PLOS ONE. After careful consideration, we feel that it has merit but does not fully meet PLOS ONE’s publication criteria as it currently stands. Therefore, we invite you to submit a revised version of the manuscript that addresses the points raised during the review process.

The manuscript focuses on important aspects of maternal and child health. However, based on the reviewers' comments, it needs major improvements. Special attention should be given to review the study design and methods. Please follow all reviewers' comments to review the manuscript. I recommend adding the  the following information in the methods section: period of data collection, coverage of the total deliveries in the same period and procedures for data quality control. 

We would appreciate receiving your revised manuscript by Dec 26 2019 11:59PM. To enhance the reproducibility of your results, we recommend that if applicable you deposit your laboratory protocols in protocols.io, where a protocol can be assigned its own identifier (DOI) such that it can be cited independently in the future. For instructions see: http://journals.plos.org/plosone/s/submission-guidelines#loc-laboratory-protocols

We look forward to receiving your revised manuscript.

Kind regards,

Marly A. Cardoso, Ph.D.

Academic Editor

PLOS ONE

Additional Editor Comments (if provided):

The manuscript focuses on important aspects of maternal and child health. However, based on the reviewers' comments, it needs major improvements. Special attention should be given to review the study design and methods. Please follow all reviewers' comments to review the manuscript. I recommend adding the the following information in the methods section: period of data collection, coverage of the total deliveries in the same period and procedures for data quality control.

Journal Requirements:

2) Please provide additional details regarding participant consent. If the need for consent was waived by the ethics committee, please include this information. If the data was accessed anonymously the lack of waiver will be acceptable.

3) Please amend either the abstract on the online submission form (via Edit Submission) or the abstract in the manuscript so that they are identical.

Reviewers' comments:

Reviewer's Responses to Questions

**Comments to the Author**

1. Is the manuscript technically sound, and do the data support the conclusions?

Reviewer #1: Partly

Reviewer #2: Yes

2. Has the statistical analysis been performed appropriately and rigorously? 

Reviewer #1: Yes

Reviewer #2: I Don't Know

3. Have the authors made all data underlying the findings in their manuscript fully available?

Reviewer #1: Yes

Reviewer #2: Yes

4. Is the manuscript presented in an intelligible fashion and written in standard English?

Reviewer #1: No

Reviewer #2: Yes

5. Review Comments to the Author

Reviewer #1: The present study has important findings and highlights the need of attention to heath conditions in LMIC, especially when considering obstetric care as a window of opportunity to social and economic development of a nation. In addition, the results signalize current alarming health issues, such as the presence of genital mutilation and high rates of stillbirth, revealing a gap in public politics in specific settings. Also, the discussion of possible connections between the stillbirth rate decrease and health care improvements reinforces the importance and the effectiveness of investments in health care.

The manuscript is well written overall. Yet, the language quality should be verify, since punctuation marks issues can be identified over the manuscript, compromising the structure, organization of the text, leading to ambiguity, doubts and demanding extra attention to interpret the information.

The high quality methodology used in the present study contributes to the credibility of the results. Both statistical analysis and data collection are detailed and well described in the text, listing the different variables and criteria considered to elaborate the GEE regression models. However, I recommend a revision of some topics before the manuscript can proceed due to some minor issues.

The introduction section brings great information about the topic and provides a contextualization of the subject to the reader. However, the given fact that “the rates of decline for South-East Asia and Africa are still very low” (page 5, 18th line) could be presented in a more reliable and solid manner if the “very low” numbers were introduced, according to the reference indicated.

At the beginning of the study setting section (page 7, 1st line), a retrospective study (Oduro et al, 2006) is used as a reference in order to describe the Navrongo War Memorial Hospital (WMH)’s services - emergency obstetric care, surgical and other public health services. However, the referred paper does not describe it, stating only that “WMH is a district health facility that offers secondary clinical and public health services” (Oduro, 2006). In addition, in the same paragraph (page 7, 10th line) the writer state that “The annual Antenatal attendance is about 2,500 and annual deliveries are hovering around 1,500”. Meantime, the paper to which this sentence refers to (Nonterah et al, 2019) informs that this data was collected from an unpublished WMH material. Thus, it would be preferable if this same consideration was made in the present study, since this material is not available to the external scientific community.

The applied definition of stillbirth as “any foetal loss after 28 weeks of gestation, and or a weight of 1000g (1kg)” (data analysis section – page 8, 1st line) is referring to a Lancet Global Health’s article (Blencowe et al, 2016). However, this is not the primary source of the stillbirth definition, but yes the International Classification of Diseases 10th revision (ICD-10). Also, according to the ICD-10, there is a third criterium to classify stillbirth: fetus of 35cm of length or more. If the writer decides not to include it, it is possible to justify the reasons for it (eg. absence of length data).

The table 1 is well presented but contains some minor issues. The variable “perineum” may not be self-explicative, demanding a more descriptive nomenclature, such as “perineum status”. Still in the table 1, the variable “maternal parity” presents 3 categories: P0, P1-P5 and >P5. Despite the explanation about the variable in the method section, the categories could be presented in a clearer manner if the 3 nominal classifications were included: primiparous, multiparous and grand multiparous. Also, the variable “gestational age” is presented with 3 categories: <37 weeks, 37 to 40 weeks and >40 weeks. However, the inclusion of the classification pre term and term (as previous presented in the text), with the corresponding intervals, may contribute to data understanding.

The results from the study are described in a very fine way. However, there are some important considerations in order to contribute to the quality of this section. The phrase “The proportion of teenage pregnancies (≤19 years) was relatively high at about 15.1%” (6th line, page 12) may be reformulated to follow the results section requirements. The use of expressions such as “high” is considered an interpretation, what is an attribute of the discussion section. In addition, to discuss if it is a high or low rate, it is important to compare the values with national/international references.

The discussion section highlights important previous findings, intertwining the existing literature and the risk factors associated with stillbirths from the presented study. However, in the discussion stablished about the C-section rate over the study period may be reconsidered in some aspects (page 18, 31st line). First, according to the text, the C-section rate of 12,5% is within the rate interval proposed by WHO (10 to 15%). Therefore, it may be considered as a positive result, providing discussions about the conditions of the Spontaneous Vaginal Delivery instead, which also requires skilled health personnel, good sanitary conditions and a satisfactory infrastructure environment. Second, the writers compare the C-section rate to the developed “world”, considering this last one as a “positive” parameter. However, developed countries usually present high C-section rates not only because of emergency obstetric care, but also due to many other factors that lead to unnecessary recommendations of this surgical procedure (eg. private health insurance interests; convenience of choosing the day of delivery). Therefore, when discussing about this specific result, it is important to conduct a careful discussion, considering all long-term benefits of the vaginal delivery.

The conclusions were drawn appropriately based on the presented data, which are fully available in the manuscript over the text, but also in tables and graphics. However, the affirmation “First time pregnant mothers and those carrying male foetus should strictly follow antenatal and prenatal services and recommendations as they are at a higher risk of stillbirths” may be reconsidered. Despite the study shows great findings, it is a retrospective e non controlled study, carrying some important limitations. Even though the literature supports many of the findings, it is essential to analyze carefully all results. For example, the correlation between pregnancies of boys and stillbirth can suggest a fact but it is not correct to stablish strong recommendations about it without a deep interpretation. The results from scientific studies may imply early actions and miss interpretations, generating negative consequences. Therefore, I suggest that the conclusion section be revised and based on solid evidenced and scientific findings.

Reviewer #2: The background section is well written. Authors started with high rates of stillbirths in the world. The authors explained how stillbirths are associated with poor health services and the availability of obstetric care services.

I will suggest updating reference 1. In line 13 of the background, I suggest abbreviating the word “Sub-Saharan Africa” in the first moment that it was presented. Also, through the paper, the authors used many abbreviations that they did not use after, I suggest to remove and only keep the abbreviations that have more the one citation.

The method section needs more work.

I think it was not clear if the author used a cross-sectional or longitudinal data. I understood that it was a cross-section study using trends.

I think would be important to describe more about the “systematically documented information”, on the second and third line of the data collection section.

Also, I would like to know more information about the “A structured designed data capturing tool”. Did the authors use a tool? Which tool was that?

The author did not specify which period the data was collected, this information should be clear.

The data was double coded but what is the reliability of the data? Could the author present a statistic measure for that?

In the analyzes section, I notice that in the GEE analyzes the authors used sometimes the groups of high risk as a reference, for example, mothers age “12-19 yrs”, mother occupation “Unemployed”, gestation >37 weeks and sometimes groups with fewer risks as normal birth weight, my suggestion is using the groups with less risk as reference “1” and establish a pattern in the analyzes.

Also, the author described what GEE means but they did not describe the assumption of the GEE.

I would like also the authors to discuss more the result's implications and recommendations.

In the last paragraph of the discussion they mentioned that the data is longitudinal, I am confused about that. Do you follow the patients between 2003-2013? if you did follow, this information should be more clear in the method section.

Please review the references for typos.

6. PLOS authors have the option to publish the peer review history of their article (what does this mean?). If published, this will include your full peer review and any attached files.

Reviewer #1: Yes: Isabel Giacomini Marques

Reviewer #2: No

---

## [Author Response · Author response to Decision Letter 0]

26 Nov 2019

Dear Sir/Madam,

Response to reviewers: Trends and risk factors associated with stillbirths: A case study of the Navrongo War Memorial Hospital in Northern Ghana (PONE-D-19-24698)

We wish to thank the academic Editor and the Reviewers for the due diligence with our manuscript. 

We do appreciate and acknowledge the comments which we believe will indeed improve the content of our paper. 

Below is a point-by-point response to the editor’s and reviewer’s comments. Also attached is the tracked and clean version for your review

It is our fervent hope that you would consider the revised manuscript suitable for the wider reader population

Sincerely

Dr Engelbert A. Nonterah. MD, MSc. 

Response to Editor’s Comments 

Editors comments Response

Please ensure that your manuscript meets PLOS ONE's style requirements, including those for file naming. The PLOS ONE style templates can be found at: We appreciate this important reminder by the editor and have now ensured that the entire manuscripts conforms to PLOS ONE style requirements 

Please provide additional details regarding participant consent. If the need for consent was waived by the ethics committee, please include this information. If the data was accessed anonymously the lack of waiver will be acceptable: Thanks to the editor for this point. We have now clarified this in the below statement: “Because secondary data was used, informed consent was waived. Further to this, we ensured confidentiality and anonymity in data extraction, processing/cleaning and analyses.”

Please amend either the abstract on the online submission form (via Edit Submission) or the abstract in the manuscript so that they are identical.: We will do just this to ensure the abstract in the manuscript is identical to the online submission form

Response to Reviewers Comments 

Reviewer #1

Comments Response

The manuscript is well written overall. Yet, the language quality should be verify, since punctuation marks issues can be identified over the manuscript, compromising the structure, organization of the text, leading to ambiguity, doubts and demanding extra attention to interpret the information.: We have edited the entire manuscript and to improve the language quality. We have also used “grammar check” to improve the manuscript where necessary

The high quality methodology used in the present study contributes to the credibility of the results. Both statistical analysis and data collection are detailed and well described in the text, listing the different variables and criteria considered to elaborate the GEE regression models. However, I recommend a revision of some topics before the manuscript can proceed due to some minor issues.: We acknowledge and appreciate the reviewers kind comments

The introduction section brings great information about the topic and provides a contextualization of the subject to the reader. However, the given fact that “the rates of decline for South-East Asia and Africa are still very low” (page 5, 18th line) could be presented in a more reliable and solid manner if the “very low” numbers were introduced, according to the reference indicated.: We have now revised this statement to read as “Saleem et al (2018) observed a 3.0% annual decline in stillbirth rate globally with much lower rates of decline for countries in South-East Asia and Africa. This the author observed will further result in the inability of these countries to achieve Every Newborn Action Plan goal of 12 per 1000births by 2030” this can be found in page 5 lines 91-95

At the beginning of the study setting section (page 7, 1st line), a retrospective study (Oduro et al, 2006) is used as a reference in order to describe the Navrongo War Memorial Hospital (WMH)’s services - emergency obstetric care, surgical and other public health services. However, the referred paper does not describe it, stating only that “WMH is a district health facility that offers secondary clinical and public health services” (Oduro, 2006). In addition, in the same paragraph (page 7, 10th line) the writer state that “The annual Antenatal attendance is about 2,500 and annual deliveries are hovering around 1,500”. Meantime, the paper to which this sentence refers to (Nonterah et al, 2019) informs that this data was collected from an unpublished WMH material. Thus, it would be preferable if this same consideration was made in the present study, since this material is not available to the external scientific community.: Thanks to the Reviewer for this observation. In our paper we have sought to expand or explain what Oduro et al 2006 meant by “secondary clinical and public health services”. These services include emergency obstetric care, surgical and other public health services as we observed. We were of the view that since we have captured this in a previous paper, it suffices to cite this. We have however rephrased this as suggested by the Reviewer. “Unpublished data from the hospital shows an average yearly Out Patient Department (OPD) attendance of about 55,000 with an average of 10,000 in-patient admissions of which maternal related conditions form part of the top ten causes of admissions. The annual Antenatal attendance is about 2,500 and annual deliveries are hovering around 1,500”

The applied definition of stillbirth as “any foetal loss after 28 weeks of gestation, and or a weight of 1000g (1kg)” (data analysis section – page 8, 1st line) is referring to a Lancet Global Health’s article (Blencowe et al, 2016). However, this is not the primary source of the stillbirth definition, but yes the International Classification of Diseases 10th revision (ICD-10). Also, according to the ICD-10, there is a third criterium to classify stillbirth: fetus of 35cm of length or more. If the writer decides not to include it, it is possible to justify the reasons for it (eg. absence of length data).: We appreciate the observation made by the Reviewer. We do agree that the primary source is ICD-10 and we have included this as a reference now. We have also justified the reason for non-exclusion of the third component of the definition in the following sentence: “The ICD-10 further states the foetus must be 35cm of length or more [23]. We however had no information on the foetal length at birth”. This can be found in page 9 lines 173-174.

The table 1 is well presented but contains some minor issues. The variable “perineum” may not be self-explicative, demanding a more descriptive nomenclature, such as “perineum status”. : Perineum corrected to “State of perineum”

Still in the table 1, the variable “maternal parity” presents 3 categories: P0, P1-P5 and >P5. Despite the explanation about the variable in the method section, the categories could be presented in a clearer manner if the 3 nominal classifications were included: primiparous, multiparous and grand multiparous. Also, the variable “gestational age” is presented with 3 categories: <37 weeks, 37 to 40 weeks and >40 weeks. However, the inclusion of the classification pre term and term (as previous presented in the text), with the corresponding intervals, may contribute to data understanding.: We have now amended this using the nominal classifications proposed by the Reviewer. “Primiparous” “multiparous” and “Grand multiparous”. We have similarly amended gestational age as Preterm (<37 weeks), Term (38-40 weeks) and Post-term (>40 weeks)

The results from the study are described in a very fine way. However, there are some important considerations in order to contribute to the quality of this section. The phrase “The proportion of teenage pregnancies (≤19 years) was relatively high at about 15.1%” (6th line, page 12) may be reformulated to follow the results section requirements. The use of expressions such as “high” is considered an interpretation, what is an attribute of the discussion section. In addition, to discuss if it is a high or low rate, it is important to compare the values with national/international references.: We indeed appreciate this observation and have made these amendments: The phrase “The proportion of teenage pregnancies (≤19 years) was relatively high at about 15.1%” has been edited to read “About 15.1% of the participants were less than 20 years of age denoting teenage pregnancy”

The discussion section highlights important previous findings, intertwining the existing literature and the risk factors associated with stillbirths from the presented study. However, in the discussion stablished about the C-section rate over the study period may be reconsidered in some aspects (page 18, 31st line). First, according to the text, the C-section rate of 12.5% is within the rate interval proposed by WHO (10 to 15%). Therefore, it may be considered as a positive result, providing discussions about the conditions of the Spontaneous Vaginal Delivery instead, which also requires skilled health personnel, good sanitary conditions and a satisfactory infrastructure environment. Second, the writers compare the C-section rate to the developed “world”, considering this last one as a “positive” parameter. However, developed countries usually present high C-section rates not only because of emergency obstetric care, but also due to many other factors that lead to unnecessary recommendations of this surgical procedure (eg. private health insurance interests; convenience of choosing the day of delivery). Therefore, when discussing about this specific result, it is important to conduct a careful discussion, considering all long-term benefits of the vaginal delivery.: We do appreciate the further explanation given by the Reviewer. We do not want to overemphasize the discussion on CS rates since the focus of the paper is stillbirth. We do agree with the Reviewer that there are several other reasons for high CS rates especially in high income countries. 

The conclusions were drawn appropriately based on the presented data, which are fully available in the manuscript over the text, but also in tables and graphics. However, the affirmation “First time pregnant mothers and those carrying male foetus should strictly follow antenatal and prenatal services and recommendations as they are at a higher risk of stillbirths” may be reconsidered. Despite the study shows great findings, it is a retrospective non controlled study, carrying some important limitations. Even though the literature supports many of the findings, it is essential to analyze carefully all results. For example, the correlation between pregnancies of boys and stillbirth can suggest a fact but it is not correct to stablish strong recommendations about it without a deep interpretation. The results from scientific studies may imply early actions and miss interpretations, generating negative consequences. Therefore, I suggest that the conclusion section be revised and based on solid evidenced and scientific findings.: Thanks for this important point. We have now edited the conclusion and it reads: “The study showed modest decline in stillbirth rate over the study period. It is evident from the study that stillbirth rate is still relatively high and some neonatal and maternal characteristics are major contributors. Efforts aimed at impacting on stillbirths should include the elimination of outmoded cultural practices such as FGM. Health systems strengthening especially in the utilization of antenatal care services and skilled attendants at delivery are essential in improving foeto-maternal outcomes”.

Reviewer #2

I will suggest updating reference 1.: We believe the information contained in the reference is still relevant but we have added a current reference to it.

In line 13 of the background, I suggest abbreviating the word “Sub-Saharan Africa” in the first moment that it was presented.: We have now corrected this accordingly

Also, through the paper, the authors used many abbreviations that they did not use after, I suggest removing and only keeping the abbreviations that have more the one citation.: We have screened through and removed the abbreviations such as KNM, OPD, VA which were not repeated in the manuscript

I think it was not clear if the author used a cross-sectional or longitudinal data. I understood that it was a cross-section study using trends. Data were collected in serial cross-sectional manner

I think would be important to describe more about the “systematically documented information”, on the second and third line of the data collection section.: We have now expanded on this to read as: “The hospital keeps record books that contain systematically documented information on each patient attending the hospital. These hospital record books includes a maternal register that is used to capture information on all maternal related cases, information captured includes maternal demographic characteristics, mode of delivery, delivery outcome (live, fresh and macerated stillbirths and birth weight), gestation at delivery, genital mutilation status of the mother”

Also, I would like to know more information about the “A structured designed data capturing tool”. Did the authors use a tool? Which tool was that? This was a designed excel spread sheet

The author did not specify which period the data was collected, this information should be clear. We have now clarified that “data were collected between 2003 and 2013”

The data was double coded but what is the reliability of the data? Could the author present a statistic measure for that? We merged data and there was a complete match hence data were reliable

In the analyzes section, I notice that in the GEE analyzes the authors used sometimes the groups of high risk as a reference, for example, mothers age “12-19 yrs”, mother occupation “Unemployed”, gestation >37 weeks and sometimes groups with fewer risks as normal birth weight, my suggestion is using the groups with less risk as reference “1” and establish a pattern in the analyzes. We have re-analysed this. For age “20-34 years” is now the reference group while for gestational age “38-40 weeks (term)” is now the reference group

Also, the author described what GEE means but they did not describe the assumption of the GEE. We have now modified the below statement to capture the relevant assumptions for GEE that relates to our analyses.: “The GEE formulated by Liang and Zeger (1986) was found to be the best regression approach for estimating the relationship of factors associated with stillbirth because of the correlation in our data which is an important assumption for using GEE [25]”

I would like also the authors to discuss more the result's implications and recommendations. We have expanded the conclusion to include recommendations:

“The study showed modest decline in stillbirth rate over the study period. It is evidence from the study that stillbirth rate is still relatively high and some neonatal and maternal characteristics are major contributors. Efforts aimed at impacting on stillbirths should include the elimination of outmoded cultural practices such as FGM. Health systems strengthening especially in the utilization of antenatal care services and skilled attendants at delivery are essential in improving foeto-maternal outcomes”.

In the last paragraph of the discussion they mentioned that the data is longitudinal, I am confused about that. Do you follow the patients between 2003 and 2013? If you did follow, this information should be clearer in the method section. We have now clarified the cross-sectional nature of our study

Please review the references for typos. Thanks for the comment. We have now revised the relevant references thoroughly

---

## [Decision Letter · Decision Letter 1]

27 Dec 2019

PONE-D-19-24698R1

Trends and risk factors associated with stillbirths: A case study of the Navrongo War Memorial Hospital in Northern Ghana

PLOS ONE

Dear Dr Nonterah,

Thank you for submitting your manuscript to PLOS ONE. After careful consideration, we feel that it has merit but does not fully meet PLOS ONE’s publication criteria as it currently stands. Therefore, we invite you to submit a revised version of the manuscript that addresses the points raised during the review process.

The authors have made major improvements in the manuscript. However, there are minor points for revision based on reviewers´ recommendations. 

We would appreciate receiving your revised manuscript by Feb 10 2020 11:59PM. To enhance the reproducibility of your results, we recommend that if applicable you deposit your laboratory protocols in protocols.io, where a protocol can be assigned its own identifier (DOI) such that it can be cited independently in the future. For instructions see: http://journals.plos.org/plosone/s/submission-guidelines#loc-laboratory-protocols

We look forward to receiving your revised manuscript.

Kind regards,

Marly A. Cardoso, Ph.D.

Academic Editor

PLOS ONE

Reviewers' comments:

Reviewer's Responses to Questions

**Comments to the Author**

1. If the authors have adequately addressed your comments raised in a previous round of review and you feel that this manuscript is now acceptable for publication, you may indicate that here to bypass the “Comments to the Author” section, enter your conflict of interest statement in the “Confidential to Editor” section, and submit your "Accept" recommendation.

Reviewer #1: All comments have been addressed

Reviewer #2: All comments have been addressed

2. Is the manuscript technically sound, and do the data support the conclusions?

Reviewer #1: Yes

Reviewer #2: Yes

3. Has the statistical analysis been performed appropriately and rigorously? 

Reviewer #1: Yes

Reviewer #2: Yes

4. Have the authors made all data underlying the findings in their manuscript fully available?

Reviewer #1: Yes

Reviewer #2: Yes

5. Is the manuscript presented in an intelligible fashion and written in standard English?

Reviewer #1: Yes

Reviewer #2: Yes

6. Review Comments to the Author

Reviewer #1: Thank you for all the responses for each one of the comments. There was a significant improvement of the manuscript, demonstrating the effort of the authors in order to develop a high quality research. However, I still have two considerations about the text.

First, I understand that the discussion about C-section is not the focus of this study. Yet, if the results about it are going to be included in the discussion, it is extremely important to conduct it carefully. In the line 304 you say that the C-section rate are within the 10-15% proposed by the World Health Organization. But then, in the next phrase, you make a statement about "This low C-section rate (...)". The sentence is not consistent since this "low" classification should not be based on developed countries rates. These countries, as I have already said in the previous review, have high C-section rates because of other elements and not only because of adequate emergency obstetric care services. Therefore, I reinforce that this discussion may be revised since C-section contributes to many short and long term negative effects in a child's life.

Second, the results in this study show the urgent need of female adequate care, mainly when discussing about FGM. Therefore, the conclusions do not show the potential of the results. I suggest that instead of the sentence "some neonatal and maternal characteristics are major contributors", the authors state the main findings, identifying these specific characteristics that contribute to neonatal and maternal health outcomes.

Overall, the quality of the writing improved, making the reading process more clear and less unambiguous.

Reviewer #2: The authors incorporated most of the suggestions and the article had an improvement. My suggestion is a review to fix minor errors.

Inline 92, there is a typo " 1000births" that need a space between the words.

Inline 106 the authors write an abbreviation for "Community-based Health and Planning Services (CHPS)" but they did not use this after through the text. I strongly recommend only to use abbreviations for words that are important for the text.

Inline 125 the authors wrote an abbreviation for "Navrongo War Memorial Hospital (WMH)", but inline 152 and 202 they used the whole name again, I recommend using only the abbreviation after the first reference in the text.

In table 1, line Estimated Gestational age (weeks), need a space between "40weeks".

In table 1, line Maternal Age (years), need a space between "< 20years".

Inline 229 they used an abbreviation for "Mother-To-Child Transmission (PMTCT)", as the authors did not use this abbreviation anymore through the text I suggest removing the abbreviation.

Please, check reference again for typos. For example, in the reference number 19 there are some dots after the first name of the author, different from the other references.

Thank you.

7. PLOS authors have the option to publish the peer review history of their article (what does this mean?). If published, this will include your full peer review and any attached files.

Reviewer #1: Yes: Isabel Giacomini Marques

Reviewer #2: No

---

## [Author Response · Author response to Decision Letter 1]

27 Dec 2019

Navrongo Health Research Centre

Ghana Health Service

P. O. Box 114

Navrongo

Mob:+233 (0) 505 989 986

E-mail: engelbert.nonterah@navrongo-hrc.org

Alternate email: drenanonterah@gmail.com

Skype: engelbert.nonterah4

 December 27, 2019

Dear Sir/Madam,

Response to reviewers: Trends and risk factors associated with stillbirths: A case study of the Navrongo War Memorial Hospital in Northern Ghana (PONE-D-19-24698R1)

We wish to thank the Reviewers for the second round of review and the comments. 

We do appreciate and acknowledge the comments as useful to improve the content of our paper. 

Below is a point-by-point response to reviewer’s comments. Also attached is the tracked and clean version for your review

It is our fervent hope that you would consider the revised manuscript suitable for publication

Sincerely

Dr Engelbert A. Nonterah. MD, MSc. 

Response to Reviewers Comments 

Reviewer #1

First, I understand that the discussion about C-section is not the focus of this study. Yet, if the results about it are going to be included in the discussion, it is extremely important to conduct it carefully. In the line 304 you say that the C-section rate are within the 10-15% proposed by the World Health Organization. But then, in the next phrase, you make a statement about "This low C-section rate (...)". The sentence is not consistent since this "low" classification should not be based on developed countries rates. These countries, as I have already said in the previous review, have high C-section rates because of other elements and not only because of adequate emergency obstetric care services. Therefore, I reinforce that this discussion may be revised since C-section contributes to many short and long term negative effects in a child's life. 

We have now deleted this statement to avoid further ambiguity and inconsistency. “This low C-section rate in the study setting may be a reflection of the inadequate emergency obstetric care services including the lack of skilled health personnel to provide such services. Therefore, this may increase in future when access to emergency obstetric care and skilled personnel improves. Presently however, the increased institutional deliveries and obstetric referrals have seen no direct effect on C-section rate across the study area. Besides the study setting being rural, other prevailing factors that might account for the low rate as some observation seem to indicate, is the fear of caesarean section by the resident population and hence the preference for home delivery”.

Second, the results in this study show the urgent need of female adequate care, mainly when discussing about FGM. Therefore, the conclusions do not show the potential of the results. I suggest that instead of the sentence "some neonatal and maternal characteristics are major contributors", the authors state the main findings, identifying these specific characteristics that contribute to neonatal and maternal health outcomes. 

We have now revised the conclusion by specifying the factors that contribute to stillbirths. The new sentence reads “Primiparous women and preterm deliveries leading to low birth weight are identified factors that result in increased stillbirths”.

Reviewer #2

Inline 92, there is a typo “1000births" that need a space between the words. We have now corrected “1000births” to “1000 births”.

Inline 106 the authors write an abbreviation for "Community-based Health and Planning Services (CHPS)" but they did not use this after through the text. I strongly recommend only using abbreviations for words that are important for the text: CHPS was used once in line 289 but we have decided to deleted the abbreviated “CHPS” and maintained the full name “Community-based Health and Planning Services” in both instances (line 106 and 289)

Inline 125 the authors wrote an abbreviation for "Navrongo War Memorial Hospital (WMH)", but inline 152 and 202 they used the whole name again, I recommend using only the abbreviation after the first reference in the text: We have now used the abbreviation “WMH” in lines 126, 152 and 202

In table 1, line Estimated Gestational age (weeks), need a space between "40weeks": We have now corrected “40weeks” to “40 weeks”.

In table 1, line Maternal Age (years), need a space between "< 20years": We have now corrected “<20years” to “< 20 years births”.

Inline 229 they used an abbreviation for "Mother-To-Child Transmission (PMTCT)", as the authors did not use this abbreviation anymore through the text I suggest removing the abbreviation. We have now deleted “PMTCT”

Please, check reference again for typos. For example, in the reference number 19 there are some dots after the first name of the author, different from the other references: The dots after the first name has been deleted and reference 19 now reads “Oduro A., Ansah P, Hodgson A, Afful T, Baiden F, Adongo P, et al. Trends in the prevalence of female genital muti-lation and its effect on delivery outcomes in the kassena-nankana district of northern Ghana. Ghana Med J. 2006;40. doi:10.4314/gmj.v40i3.55258”.

We have checked other references: References 11, 14, 15 have all been edited

---

## [Editor Report · Decision Letter 2]

29 Jan 2020

Trends and risk factors associated with stillbirths: A case study of the Navrongo War Memorial Hospital in Northern Ghana

PONE-D-19-24698R2

Dear Dr. Nonterah,

We are pleased to inform you that your manuscript has been judged scientifically suitable for publication and will be formally accepted for publication once it complies with all outstanding technical requirements.

With kind regards,

Marly A. Cardoso, Ph.D.

Academic Editor

PLOS ONE
---

## [Editor Report · Acceptance letter]

10 Feb 2020

PONE-D-19-24698R2 

Trends and risk factors associated with stillbirths: A case study of the Navrongo War Memorial Hospital in Northern Ghana 

Dear Dr. Nonterah:

I am pleased to inform you that your manuscript has been deemed suitable for publication in PLOS ONE. Congratulations! Your manuscript is now with our production department. 

With kind regards,

on behalf of

Dr. Marly A. Cardoso 

Academic Editor

PLOS ONE